# Optimising Artificial Moss Growth for Environmental Studies in the Mediterranean Area

**DOI:** 10.3390/plants10112523

**Published:** 2021-11-19

**Authors:** Zulema Varela, Carlos Real, Cristina Branquinho, Teresa Afonso do Paço, Ricardo Cruz de Carvalho

**Affiliations:** 1CRETUS, Ecology Unit, Department Functional Biology, Facultade de Bioloxía, Universidade de Santiago de Compostela, 15782 Santiago de Compostela, Spain; 2Centre for Ecology, Evolution and Environmental Changes (cE3c), Faculdade de Ciências, University of Lisbon, 1749-016 Lisbon, Portugal; cmbranquinho@fc.ul.pt (C.B.); rfcruz@fc.ul.pt (R.C.d.C.); 3Ecology Unit, Department Functional Biology, Escola Politécnica Superior de Enxeñaría (EPSE), Universidade de Santiago de Compostela, Rúa Benigno Ledo, Campus Terra, 27002 Lugo, Spain; carlos.real@usc.es; 4LEAF—Linking Landscape, Environment, Agriculture and Food—Research Center, Associated Laboratory TERRA, Instituto Superior de Agronomia, Universidade de Lisboa, Tapada da Ajuda, 1349-017 Lisboa, Portugal; tapaco@isa.ulisboa.pt; 5Marine and Environmental Sciences Centre (MARE), Faculdade de Ciências, Universidade de Lisboa, Campo Grande, Edifício C2, Piso 5, 1749-016 Lisboa, Portugal

**Keywords:** bryophytes, ecological restoration, green roofs, Moss cover, photoperiod, temperature

## Abstract

Bryophytes are poikilohydric organisms that play a key role in ecosystems, while some of them are also resistant to drought and environmental disturbances but present a slow growth rate. Moss culture in the laboratory can be a very useful tool for ecological restoration or the development of urban green spaces (roof and wall) in the Mediterranean region. Therefore, we aim to: (i) determine the optimal culture conditions for the growth of four moss species present in the Mediterranean climate, such as *Bryum argenteum*, *Hypnum cupressiforme*, *Tortella nitida,* and *Tortella squarrosa*; (ii) study the optimal growth conditions of the invasive moss *Campylopus introflexus* to find out if it can be a threat to native species. Photoperiod does not seem to cause any recognisable pattern in moss growth. However, temperature produces more linear but slower growth at 15 °C than at 20 and 25 °C. In addition, the lower temperature produced faster maximum cover values within 5–8 weeks, with at least 60% of the culture area covered. The study concludes that the culture of moss artificially in the organic gardening substrate without fertilisers is feasible and could be of great help for further use in environmental projects to restore degraded ecosystems or to facilitate urban green spaces in the Mediterranean area. Moreover, this study concludes that *C. introflexus* could successfully occupy the niche of other native moss species, especially in degraded areas, in a future global change scenario.

## 1. Introduction

Mosses play an important role in ecosystems. They help improve soil stability, fix basic nutrients like nitrogen (N) and carbon (C), and increase the organic matter content in soils, facilitating other plants to grow roots and serving as a habitat for other organisms [1]. They are also susceptible to rapid dehydration under low relative humidity and quickly resume metabolic activity upon rehydration [2,3,4]. Furthermore, mosses can recover quickly after an environmental perturbation, being one of the earlier colonizers among biological soil crust (BSC) components [5]. Moreover, they are totipotent, i.e., any vegetative moss tissue can be a propagule from which grows a new plant [6]. In addition, due to their poikilohydric nature, they have a great ability to capture and retain atmospheric pollutants and, therefore, are widely used to monitor air quality [7,8].

All these characteristics make mosses the ideal organisms for ecological restoration projects in disturbed ecosystems [9,10] or even for use on roofs and green walls in urban environments [11]. However, is to not acceptable to harvest mosses in the field and transplant them elsewhere, as their growth rate in the natural environment is very slow. For example, Giquan et al. [12] report that moss *Bryum argenteum* is able to cover 70% of 10 × 10 cm squares in 3–4 years in desert dunes, so another source of mosses is needed, which is why laboratory cultures were started. Studies aimed at restoring or rehabilitating dryland areas have been artificially growing moss for years [10,12,13,14,15,16,17,18,19,20,21]. The procedure followed by most of these works is practically the same: they collect bryophytes of BSCs from the study area and grow them in greenhouses or growing chambers inside plastic containers using autoclaved sand from such areas as a substrate and adding culture medium to speed up their growth. However, these moss culture methods have been poorly described and no information is available concerning their performance. It is unknown which photoperiod or temperature they use to cultivate the different species or if the moss samples were irrigated for most studies. It is only noted that high rates of moss growth are reached in two and six months.

However, there is scarce previous experience of artificial moss culture in non-arid environments in Europe, unlike in North America or Asia for gardening and even moss graffiti [22,23]. Although there are many arid areas in the Mediterranean area, plenty of other areas have a temperate climate and, therefore, other types of ecosystems and other moss species adapted to these conditions. According to Proctor [24], the main growth period for most bryophytes in temperate climates with uniform rainfall throughout the year tends to be in autumn. In winter, growth is limited by low temperatures and in summer by lack of water [25,26]. Furnes and Grime [27] noted that the optimum temperature for the growth of most species is between 15 and 25 °C. Glime [28] conjectures that temperature may control when and where species germinate and thus limit their distribution. However, this author reported that each moss species will grow at a certain rate and that this will be determined by factors other than temperature, such as the photoperiod and its intensity, the species’ morphology, and its growth or life form. For example, in the moss *Ceratodon purpureus,* long days stimulate protonema elongation, while short days result in protonema branching. Meanwhile, in *Bryum pseudotriquetrum,* days of ten or more hours are necessary for protonema germination and growth. Therefore, there is undoubtedly a need to optimise the specific technique for the species of interest for cultivation.

This study aims to: (i) determine the optimal culture conditions for the following four moss species commonly found in the Mediterranean (dry summer) climate zones as a tool to facilitate ecological restoration or green roofs/walls projects: *Bryum argenteum, Hypnum cupressiforme, Tortella nitida,* and *Tortella squarrosa*. On one hand, *B. argenteum, T. nitida,* and *T. squarrosa* are acrocarpic moss species very common in the Mediterranean and quite tolerant to desiccation [29]. An example of pleurocarpic moss is *H. cupressiforme*, a cosmopolitan species which is present in every continent (except Antarctica) and spread over a wide range of habitats and climatic areas [29]. Our second aim is to (ii) study the optimal growth conditions of the invasive moss *Campylopus introflexus* to find out if it can be a threat to native species. This species is considered one of the 100 worst alien species in Europe [30]. Moreover, this moss is native to the southern hemisphere and was introduced to Europe in the 1940s [31], and its spread has been increasing with climate change.

## 2. Results

Table 1 shows the average of the initial, final, and maximum cover, and the time it took to reach this maximum cover. The variability within all treatment combinations was large (standard deviation up to 29%), even though the initial conditions were the same for all replicates. At 15 °C, the maximum cover was reached towards the end of the experiment (eight weeks) in most cases, while at 20 and 25 °C the maximum cover was usually reached between weeks 4 and 6, decreasing afterwards. It should also be noted that, except for *C. introflexus*, species reached the maximum cover at 15 °C, decreasing slightly at 20 °C and more at 25 °C.

The adjusted GAM curves (Figure 1) showed a linear growth pattern at 15 °C and parabolic growth patterns at 20 °C and 25 °C, in line with the results of the maximum coverage. The determination coefficients (r^2^) for all curves were significant (*p* < 0.05), with the higher values being mostly at 15 °C (e.g., *T. nitida* with a photoperiod of 12 h, r^2^ = 0.96), except for *H. cupressiforme* which is at 20 °C, with a photoperiod of 20/4 h (r^2^ = 0.45) and with slightly lower adjusting values.

On the other hand, a significant response of moss growth to the studied variables was not always found (see F-value in Figure 1). Thus, while *C. introflexus*, *H. cupressiforme*, and *T. nitida* showed a significant response to temperature and photoperiod, *B. argenteum* showed it to photoperiod only and *T. squarrosa* is independent of both.

## 3. Discussion

### 3.1. Moss Species Growth

The culture of different moss species in growth chambers was very satisfactory. Growth was observed for all species and all treatments, in some cases covering almost 80% of the culture area. High variability was also found, which could be caused by working with fragments of live mosses as some can grow larger than others. Nevertheless, this variability, which is impossible to predict, would not limit the moss growth technique for use in environmental studies.

According to the literature, the optimum growth for most moss species is between 15 and 25 °C [27]. In general, we observed a lower growth rate at 15 °C with a linear pattern that reached maximum coverage towards the end of the experiment (Table 1 and Figure 1). In contrast, at 20 and 25 °C, we observed that the maximum rate of growth for species is reached towards the middle of the experiment, as if they were reaching their optimal growth and from then they start to die and the cover decreases. Higher temperatures will favour higher photosynthesis and therefore a higher rate of growth [25], but due to some unidentified stress (maybe competition between different moss fragments) the mosses under these conditions start to die midway through the experiment. Regarding photoperiod, it could be expected that more hours of light would favour growth [28] and despite finding significant relations between photoperiod and the rate of growth of *B. argenteum, C. introflexus, H. cupressiforme,* and *T. nitida* (see Figure 1), the maximum cover was achieved in any of the three photoperiods without any apparent pattern. In vascular plants, the temperature is one of the key factors determining their metabolism, growth, and distribution, so it could be assumed that temperature is more limiting than photoperiod for moss growth [32].

It would be expected that acrocarpous species, such as *B. argenteum*, *T. nitida,* and *T. squarrosa*, would grow more at higher temperatures and with longer photoperiods. These smaller species with cushion life forms can equilibrate more slowly with the relative humidity of their environment and therefore resist desiccation better [4,33]. However, this was only observed for *T. nitida* where the growth achieved at 20 °C and with a 20/4 h photoperiod was much higher than for other temperatures. Although these species are resistant to desiccation, with the increase in temperature there may have been a decrease in water availability and therefore a change in growing conditions that has affected them.

In contrast, the only pleurocarpous moss in the study, *Hypnum cupressiforme*, does not seem to reach the cover levels of the acrocarpous mosses and shows more growth at lower temperatures (15 °C), except at certain times of the 20/4 h photoperiod. *H. cuppresiforme* is cosmopolitan and its life form is mats, which are more commonly found in humid and shady sites [34]. Therefore, temperatures generally above 20 °C could be limiting their growth, as observed in the pleurocarpous moss *Hylocomium splendens* whose photosynthetic activities and growth rates decreased in warming winters in subarctic areas [35].

As expected, for most species except *T. squarrosa*, temperature and photoperiod influenced their growth. In general, to achieve high coverage in a short time, growing the mosses at 20 °C and with a 12/12 h photoperiod will allow to reach high cover in five weeks. If, on the other hand, the objective is to achieve maximum cover but at a slower rate, it is better to grow at 15 °C. Consequently, the next step would be to transplant this cultured moss to degraded ecosystems, arid areas, or urban green spaces in the Mediterranean area to confirm that the moss can grow in these conditions to aid restoration.

### 3.2. Alien Invasive Species

*Campylopus introflexus* is an invasive acrocarpic moss from the Southern Hemisphere that is widespread in Europe thanks to its high ecological tolerance [36] and ability to resist in environments with limited water availability [37]. There are no plans to use this exotic species in future restoration projects. However, our results showed that *C. introflexus* could interfere with the growth of other species when these are transplanted to the field [38]. The results show that it tends to grow more at higher temperatures, e.g., with a 12/12 h photoperiod it grows more at 20 °C and 25 °C than at 15 °C and with a 20/4 h photoperiod it grows more at 25 °C (Figure 1). This means that in a possible future scenario of global change with increasing temperatures and increasing periods of drought, *C. introflexus* could successfully occupy the habitat of other native moss species, especially in degraded areas where it occurs frequently [28].

## 4. Materials and Methods

### 4.1. Moss Processing and Experimental Design

Moss species were collected from roadsides, full sun-exposed walls in dry sub-humid (Alegrete, Parque Natural de São Mamede; Barreiro, Setúbal, Portugal) to semiarid (Zebreira, Beira Baixa; Estremoz, Alto Alentejo, Portugal), from natural to urban locations (Table 2).

Samples were collected dry and stored in paper bags until use. Once in the laboratory, they were cleaned from plant remains, epiphytes, adhering soil particles, etc. and dried at room temperature (circa 20 °C). Once dried, the moss shoots were cut into smaller pieces with scissors to be used as propagules. The experiment was carried out in growth chambers (Aralab, Portugal). We filled 750 mL capacity plastic containers (length = 13.8 cm, width: 11.8 cm and height: 5.0 cm) with 150 g of the organic soil substrate commonly used for gardening (N: 150 mg/kg; P: 150 mg/kg; K: 205 mg/kg; CaCO3: 6.34 mg/kg; organic matter: 50% ± 10%; pH: 6 ± 1; Auchan) that we irrigated with 200 mL of water until all the substrate was humidified. Subsequently, we weighed different amounts of propagule according to the moss species and spread them by hand in the different containers considering 3 replicates per species and treatment. Once prepared, we maintained them under different controlled conditions of temperature (15, 20 and 25 °C) and photoperiod (12/12 h, 16/8 h, and 20/4 h, day/night). As each species has its optimum temperature and light, we decided to test between the minimum (15 °C and 12 h) and maximum range (25 °C and 20 h) that the literature reports for mosses’ growth, to establish their temperature and light preferences [24]. The humidity remained constant at 50% throughout the experiment and the samples were irrigated each week with 200 mL of tap water. The experiment was intended to last until the mosses attained a 100% cover, but after 8 weeks there was an intense growth of fungi and algae in the samples, so the experiment was terminated. No fertilisers, antifungals, etc. were added during the time of the experiment. Photos of *H. cupressiforme* at 25 °C were not included in the analysis as we were unable to differentiate the green moss pixels from the green algae that grew on these samples from the beginning of the experiment.

### 4.2. Moss Growth Measurement

To estimate the growth of the mosses in a non-destructive way, we measured their cover weekly for the eight weeks of the experiment. To do this, we took digital RGB images (Nikon D5100 camera) of each container (1134 images = 4 species ×3 replicates ×3 temperature treatments ×3 light treatments ×9 weeks + *H. cupressiforme* species ×3 replicates ×2 temperature treatments ×3 light treatments ×9 weeks). All images were taken in a well-lit area close to a window in the laboratory but avoiding direct sunlight, and with the camera set in automatic mode.

The colour differences between the substrate (dark brown/black) and the moss (various shades of green) allowed for the classification of the pixels of the images as substrate or moss, and the calculation of the percentage cover of the moss. To do this, the first step was to isolate the substrate area by colouring the other pixels in red (see Figure 2). We did this manually, using the masking routines in the Gimp Image Editor software (Berkeley, CA, USA). The second step was to classify the pixels. This was done using the functions in the R package pixelclasser [39], which allow the user to define classification rules using a set of test images containing examples of the categories to recognize. The import function of pixelclasser transforms the original RGB values into proportions (the so-called rgb colour space) which eliminates colour differences due to the intensity of illumination and simplifies the analysis of the image into a two-dimensional problem. The pixels are then plotted in the plane defined by two of the r, g, or b variables, and the user can trace one or more straight lines that serve as classifying rules for the pixels. This is a simplified, manual variant of the technique known as support vector machine [40,41]. Figure 2 illustrates the procedure in a series of three images.

### 4.3. Statistical Analysis

To characterize the growth of the different moss species and their response to temperature and photoperiod, we fitted generalized additive models (GAM) to the percentage cover data for each combination of treatments using the “gam” function of the “mgcv” R package [42] and “ggplot2” package [43] to plot them. The statistical analysis was performed in version R 3.4.3 [44].

## 5. Conclusions

The findings of this study demonstrate that it is possible to grow moss artificially in plastic containers with organic gardening substrate without fertilisers, achieving an average coverage of the culture area of more than 60% in 5–8 weeks. The growth of all species except *T. nitida* is linear and sustained at 15 °C. At other temperatures (20 °C and 25 °C), regardless of the photoperiod, growth stops, and the moss dies after some weeks. This would confirm that moss culture in the laboratory could be of great help for further use in environmental projects. Finally, in a future of global change with higher temperatures and more periods of drought, *C. introflexus* could successfully occupy the niche of other native moss species, especially in degraded areas.

## Figures and Tables

**Figure 1 plants-10-02523-f001:**
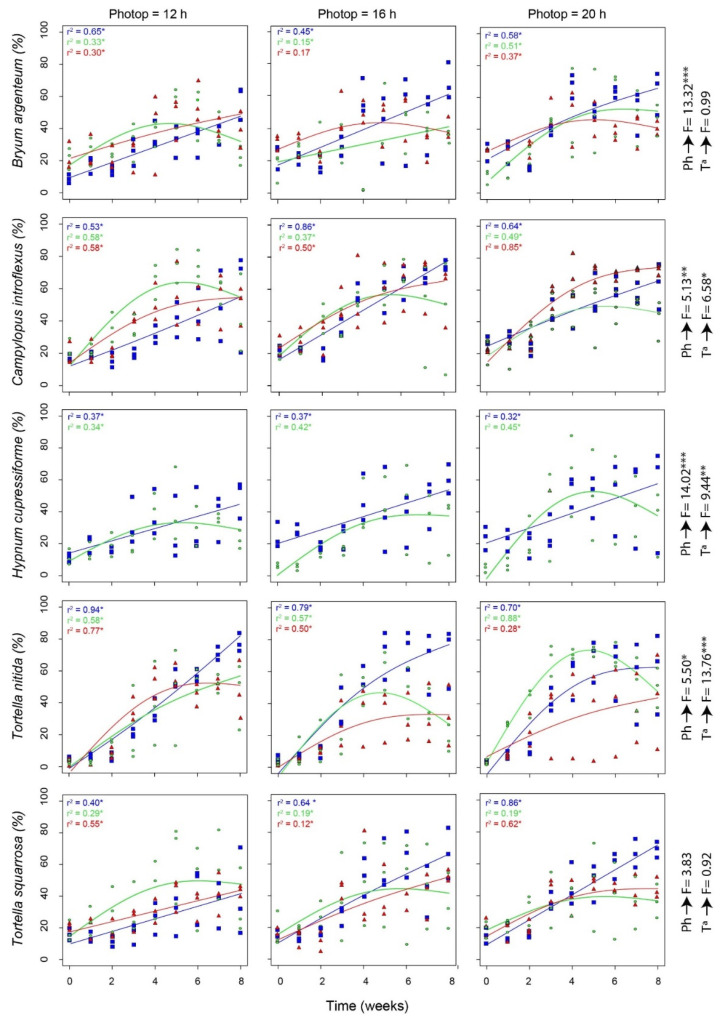
Tendency of a generalized additive model (GAM) for the coverage of the different moss species and the photoperiods and the different temperatures (Solid lines: blue = 15 °C, green = 20 °C and red = 25 °C), adjusted r^2^, F-value (F) for both variables and *p*-value (*** *p* < 0.001; ** *p* < 0.01; * *p* < 0.05).

**Figure 2 plants-10-02523-f002:**
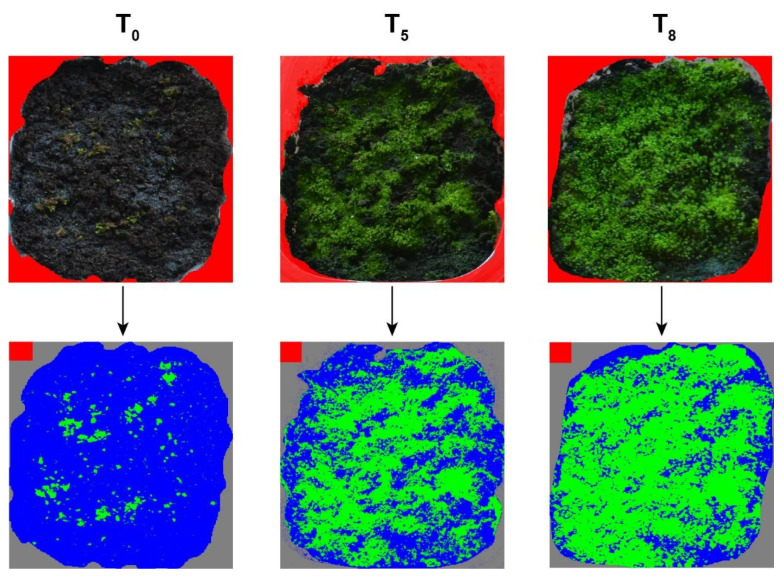
Images of *Tortella nitida* at time 0 (T_0_), 5 (T_5_) and 8 (T_8_) weeks and their corresponding pixel coverage analysis.

**Table 1 plants-10-02523-t001:** Mean of the initial, final, and maximum coverage (%) and the standard deviation of the different moss species for the different variables and time (weeks) to cover it.

Species	T (°C)	Photoperiod (Hours)	Initial Cover (%)	Final Cover (%)	Max. Cover (%)	Time to Max. Cover (Weeks)
*Bryum* *argenteum*	15	12 h	8.63 ± 2.15	58.10 ± 10.95	58.10 ± 10.95	8
		16 h	23.60 ± 7.81	69.01 ± 11.40	69.01 ± 11.40	8
		20 h	25.77 ± 5.89	64.54 ± 13.31	64.54 ± 13.31	8
	20	12 h	21.87 ± 6.73	22.78 ± 6.03	54.82 ± 12.73	5
		16 h	16.53 ± 9.05	35.43 ± 3.81	50.26 ± 18.74	5
		20 h	10.37 ± 4.67	46.38 ± 9.86	60.87 ± 21.07	6
	25	12 h	22.27± 8.45	39.13 ± 10.88	55.96 ± 12.69	6
		16 h	31.63 ± 4.33	39.95 ± 6.88	49.93 ± 7.13	4
		20 h	26.93 ± 0.44	40.92 ± 4.24	46.17 ± 10.87	4
*Campylopus* *introflexus*	15	12 h	17.09 ± 2.49	56.43 ± 31.62	56.43 ± 31.62	8
		16 h	21.24 ± 2.99	73.98 ± 3.32	73.98 ± 3.32	8
		20 h	27.56 ± 2.26	62.23 ±13.90	62.23 ±13.90	8
	20	12 h	22.04 ± 5.45	41.89 ± 24.65	74.12 ± 8.47	6
		16 h	19.68 ± 4.18	44.15 ± 35.21	69.33 ± 7.62	6
		20 h	22.40 ± 8.70	41.40 ± 12.56	52.28 ± 2.28	7
	25	12 h	18.61 ± 7.53	49.81 ± 11.68	55.47 ± 19.45	5
		16 h	24.24 ± 5.90	56.80 ± 18.46	64.48 ± 17.68	7
		20 h	21.16 ± 1.08	72.00 ± 3.20	72.47 ± 0.44	7
*Hypnum cupressiforme*	15	12 h	11.35 ± 2.12	49.24 ± 11.87	49.24 ± 11.87	8
		16 h	24.53 ± 8.11	60.46 ± 8.79	60.46 ± 8.79	8
		20 h	23.71 ± 7.33	52.40 ± 33.10	53.33 ± 9.22	4
	20	12 h	12.03 ± 4.45	22.44 ± 5.79	41.5 ±23.66	5
		16 h	6.86 ± 1.23	47.92 ± 12.27	32.93 ± 17.73	5
		20 h	4.49 ± 2.06	34.51 ± 20.34	64.08 ± 25.55	4
*Tortella nitida*	15	12 h	4.88 ± 0.75	77.89 ± 5.61	77.89 ± 5.61	8
		16 h	5.32 ± 1.99	74.30 ± 11.67	70.87 ± 17.74	6
		20 h	3.92 ± 0.35	61.46 ± 24.99	68.53 ± 13.23	5
	20	12 h	2.71 ± 0.65	46.38 ± 20.83	54.85 ± 6.26	6
		16 h	2.69 ± 2.14	18.05 ± 8.58	61.40 ± 12.93	5
		20 h	2.56 ± 0.85	50.78 ± 12.54	73.26 ± 2.65	5
	25	12 h	3.10 ± 1.90	47.40 ± 18.06	56.54 ± 7.57	5
		16 h	2.00 ± 0.71	32.10 ± 19.31	33.36 ± 16.23	5
		20 h	3.16 ± 0.47	43.02 ± 29.56	43.02 ± 29.56	8
*Tortella squarrosa*	15	12 h	15.68 ± 3.43	40.30 ± 27.24	43.52 ± 18.14	6
		16 h	15.88 ± 2.36	67.09 ± 16.27	67.09 ± 16.27	8
		20 h	14.99 ± 4.90	69.15 ± 5.06	69.15 ± 5.06	8
	20	12 h	17.21 ± 6.19	34.65 ± 20.59	64.21 ± 25.26	5
		16 h	18.03 ± 10.46	35.67 ± 17.83	49. 63 ± 26.86	5
		20 h	19.29 ± 5.13	36.49 ±10.86	47.48 ± 28.05	5
	25	12 h	20.87 ± 2.53	43.20 ± 2.82	43.20 ± 2.82	8
		16 h	15.54 ± 3.65	52.76 ± 3.00	52.76 ± 3.00	8
		20 h	20.14 ± 6.30	44.64 ± 6.66	48.15 ± 3.56	5

**Table 2 plants-10-02523-t002:** Bryophyte species collected in the present study (clade, growth form, location, aridity index (1980–2010) (A.I.), natural (N), or urban (U) site, coordinates).

Species	Plant Clade	Growth Form	Location	A.I. *	N/U	Coordinates
*Bryum argenteum* Hedw.	Bryophyta (mosses)	Acrocarpous	Zebreira	Semi-arid	U	39°51′06.9′′ N 7°04′22.9′′ W
*Campylopus introflexus* (Hedw.) Brid.	Bryophyta (mosses)	Acrocarpous	Barreiro	Dry sub-humid	N	38°36′50.5′′ N 9°02′31.9′′ W
*Hypnum cupressiforme* Hedw.	Bryophyta (mosses)	Pleurocarpous	Alegrete (Parque Natural de São Mamede)	Dry sub-humid	N	39°15′14.6′′ N 7°18′05.0′′ W
*Tortella nitida* (Lindb.) Broth.	Bryophyta (mosses)	Acrocarpous	Estremoz	Semi-arid	U	38°48′01.8′′ N 7°39′41.9′′ W
*Tortella squarrosa* (Brid.) Lindb.	Bryophyta (mosses)	Acrocarpous	Zebreira	Semi-arid	U	39°51′06.9′′ N 7°04′22.9′′ W

* Aridity Index (1980–2010) according to Kurz-Besson et al. (2016).

## Data Availability

Not applicable.

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
