# Peer review of "Optimising Artificial Moss Growth for Environmental Studies in the Mediterranean Area"

_plants, 2021, doi:10.3390/plants10112523_

Round 1

Reviewer 1 Report

The present study is intended to be a contribution to the artificial growth of some moss species. which could be a practical support for those interested in producing moss pads, which can then be used for other purposes (green roofs, etc!).

In the abstract, it is stated that these species are cultivated to be used in ecological reconstruction, which does not correspond to the discussions, where it is correctly stated that the species Campylopus intreoflexus is alien and will not be used for this purpose. For the same reason, I consider that the objectives do not correspond either (row 73-75)

In this study, the authors underline the idea that the artificially cultivated material will be used for ecological reconstruction without specifying the efficiency from the financial point of view. Moreover, the methods specify that the pursuit of the experiment was interrupted faster than reaching a 100% coverage, because fungi and algae were installed - which indicates that the experiment is not a success - contrary to the statement in row 113

The practical relevance of this study is not clear - in the Mediterranean area summer is warmer so that only Campylopus survive, actually an alien invasive species

there is no previous experience of artificial moss culture in non-arid environments – what about moss graffiti

Although the introduction specifies that the literature does not provide enough details about artificial growth, this paper does not specify the nature of the substrate - solid or liquid, soil, agar ..., thickness, size of plastic boxes, more precisely the surface of each box!

REFERENCES - some are incomplete – e.g. the first one

DeFalco, L. A., Detling, J. K., Tracy, C. R., & Warren, S. D. (2001). Physiological variation among native and exotic winter annual plants associated with microbiotic crusts in the Mojave Desert. Plant and Soil234(1), 1-14.

references are not uniformly written - sometimes years are not bolded, the use of &

Author Response

  • In the abstract, it is stated that these species are cultivated to be used in ecological reconstruction, which does not correspond to the discussions, where it is correctly stated that the species Campylopus intreoflexus is alien and will not be used for this purpose. For the same reason, I consider that the objectives do not correspond either (row 73-75)

We agree with the reviewer so we add new objective (lines 81-97) and change the abstract accordingly: “to study the optimal growth conditions of the invasive moss Campylopus introflexus to find out if it can be a threat to native species. This species is considered one of the 100 worst alien species in Europe [30], this moss is native to the southern hemisphere and was introduced to Europe in the 1940s [31], and its spread has been increasing with climate change”.

  • In this study, the authors underline the idea that the artificially cultivated material will be used for ecological reconstruction without specifying the efficiency from the financial point of view. Moreover, the methods specify that the pursuit of the experiment was interrupted faster than reaching a 100% coverage, because fungi and algae were installed - which indicates that the experiment is not a success - contrary to the statement in row 113E.

We agree with the reviewer that the financial point of view is important but believe that this is an objective of further work. With the optimal conditions for growing moss in the laboratory, the next step is to transplant the moss to the field and verify that it can grow in those conditions and that is where the economic point of view would come in.  In any case, as can be seen in the bibliography on the topic [10-21].

Regarding the fact that the experiment has not been a success… we consider that it has been a success, but we have changed the word “successful” for “very satisfactory” (line 126). In many cases, after 5 weeks, almost 80% of the culture surface is covered, but until the end of the experiment we did not know that this was the optimum growth rate under these conditions, nor that by 8 weeks we were going to have fungus and algae growth. Now we know.

  • The practical relevance of this study is not clear - in the Mediterranean area summer is warmer so that only Campylopus survive, actually an alien invasive species.

It is clear that Campylopus introflexus survives in the Mediterranean area but it is unknown whether more daylight hours and increased temperature will limit its growth or help it to become more competitive. It is a moss that likes degraded areas and may be a problem when trying to get other moss species to grow there when trying to restore the area. We believe that with the new wording of the manuscript, considering 2 different but complementary objectives, this problem disappears.

  • There is no previous experience of artificial moss culture in non-arid environments – what about moss graffiti

We agree with the reviewer and have therefore added this sentence (line 64-66): “However, there is scarce previous experience of artificial moss culture in non-arid environments in Europe, unlike in North America or Asia for gardening and even moss graffiti [22,23]”.

  • Although the introduction specifies that the literature does not provide enough details about artificial growth, this paper does not specify the nature of the substrate - solid or liquid, soil, agar ..., thickness, size of plastic boxes, more precisely the surface of each box!

In agreement with the reviewer, we have added to the text:

  • The size description of the plastic containers (Line 194 and 1895): “length = 13.8 cm, width: 11.8 cm and height: 5. 0 cm”.
  • “soil” in the description of the substrate (line 195).

  • REFERENCES - some are incomplete – e.g. the first one: DeFalco, L. A., Detling, J. K., Tracy, C. R., & Warren, S. D. (2001). Physiological variation among native and exotic winter annual plants associated with microbiotic crusts in the Mojave Desert. Plant and Soil234(1), 1-14.

references are not uniformly written - sometimes years are not bolded, the use of &

We have reviewed all the literature and it should now be uniform.

Reviewer 2 Report

I feel that the paper is well written. I have only a few suggestions.

  1. As a result, how about suggesting tables about graphs? I think it will be helpful to understand your result exactly.
  2. In methodology, how about showing a diagram about process for easy understanding?
  3. In discussion, how about using subtitles for easy finding of your paper’s originality?

Author Response

I feel that the paper is well written. I have only a few suggestions.

  1. As a result, how about suggesting tables about graphs? I think it will be helpful to understand your result exactly

I'm sorry but we don't quite understand what it means to suggest tables on top of graphs. All the values in Table 2 are in Figure 1 but we think it is easier to see the precise percentages with the table and the growth pattern with the figure.

  1. In methodology, how about showing a diagram about process for easy understanding?

The study is not too extensive and already consists of a graphical abstract, 2 tables and 2 figures, we believe it is enough. In addition, with the graphical abstract, the explanation of material and methods and the Figure 2, is enough to understand the work but if the reviewer requires another diagram, figure or photos of the process we will add them.

  1. In discussion, how about using subtitles for easy finding of your paper’s originality?

We agree with the reviewer so we’ve divided the discussion into 2 sections.

Reviewer 3 Report

The reviewed manuscript "Optimising artificial moss growth for environmental studies in the Mediterranean area" is a simple yet interesting piece of work. The authors determined the optimal culture conditions for the growth of five moss species present in the Mediterranean climate. In conclusion, photoperiod does not cause any recognisable pattern in moss growth and temperature produces linear but slower growth at 15 °C than at 20 and 25 °C. In addition, the lower temperature produced faster maximum moss growth within 5 to 8 weeks, with at least 60% of the culture area covered.

It would be interesting to compare (or at least discuss) obtained results with the growth rate of mosses in the natural environment

Author Response

The reviewed manuscript "Optimising artificial moss growth for environmental studies in the Mediterranean area" is a simple yet interesting piece of work. The authors determined the optimal culture conditions for the growth of five moss species present in the Mediterranean climate. In conclusion, photoperiod does not cause any recognisable pattern in moss growth and temperature produces linear but slower growth at 15 °C than at 20 and 25 °C. In addition, the lower temperature produced faster maximum moss growth within 5 to 8 weeks, with at least 60% of the culture area covered.

It would be interesting to compare (or at least discuss) obtained results with the growth rate of mosses in the natural environment.

We agree with the reviewer that moss growth rate values in natural environments could be added to the text to further highlight the need for laboratory cultivation if large quantities are required so we have added in the Introduccion (lines 50-52):  “as its growth rate in natural environment is very slow as for example Giquan et al. [12] report that moss Bryum argenteum is able to cover 70% of 10 x 10 cm squares in 3-4 years in desert dunes”.
